# Soybean-Oil-Based CO_2_-Switchable Surfactants with Multiple Heads

**DOI:** 10.3390/molecules26144342

**Published:** 2021-07-18

**Authors:** Huiyu Huang, Xiaoling Huang, Hongping Quan, Xin Su

**Affiliations:** 1State Key Laboratory of Polymer Materials Engineering, Polymer Research Institute, Sichuan University, Chengdu 610065, China; hyhuang31235@163.com (H.H.); huangxiaoling_hxl@163.com (X.H.); 2Oil & Gas Field Applied Chemistry Key Laboratory of Sichuan Province, School of Chemistry and Chemical Engineering, Southwest Petroleum University, Xindu 610500, China; quanhp2005@swpu.edu.cn

**Keywords:** oligomeric surfactant, CO_2_ switching, surface activity, viscosity, soybean oil, emulsion

## Abstract

Oligomeric surfactants display the novel properties of low surface activity, low critical micellar concentration and enhanced viscosity, but no CO_2_ switchable oligomeric surfactants have been developed so far. The introduction of CO_2_ can convert tertiary amine reversibly to quaternary ammonium salt, which causes switchable surface activity. In this study, epoxidized soybean oil was selected as a raw material to synthesize a CO_2_-responsive oligomeric surfactant. After addition and removal of CO_2_, the conductivity analyzing proves that the oligomeric surfactant had a good response to CO_2_ stimulation. The viscosity of the oligomeric surfactant solution increased obviously after sparging CO_2_, but returned to its initial low viscosity in the absence of CO_2_. This work is expected to open a new window for the study of bio-based CO_2_-stimulated oligomeric surfactants.

## 1. Introduction

Traditional surfactants are composed of hydrophilic and hydrophobic groups. However, due to the separation caused by the electric repulsion among the ionic head groups or the hydration effect of the ionic groups [1,2], it is difficult for these groups to arrange closely at the interface or in the molecular aggregates, thus resulting in the relatively low surface activity of the surfactants [3]. Despite the optimal solubility-enhancing properties, thickening behavior, dispersibility and flocculability possessed by the polymeric surfactants, it is generally difficult to realize a directional arrangement of the polymeric surfactant molecules at the interface [3]. Their surface tension is observed to be weaker than the traditional surfactants [4,5]. Thus, it takes much longer for these surfactants to reach the surface tension equilibrium. Thus, the existing shortcomings limit the applications of the traditional and polymeric surfactants. The advent of oligomeric surfactants has bridged the gap between the two surfactant types. Due to their various characteristics, such as high efficiency, multifunctionality and environmental friendliness, the oligomeric surfactants have recently emerged as new generation surfactants and are most likely to find a widespread use [6,7,8,9].

The oligomeric surfactants represent a special class of novel surfactants comprising two or more traditional surfactant molecules connected by chemical bonds at or near the head group through a spacer group. Based on the number of connected amphiphilic units [10], the oligomeric surfactants can be classified as dimeric (Gemini) [11,12], trimeric [13], tetrameric [14], multimeric [15], etc. The association of the surfactant molecules via chemical bonds instead of via simple physical means not only ensures the close contact among the active ingredients of the oligomeric surfactants, but also guarantees that the hydrophilic characteristics of the head groups are not destroyed, thus enabling the oligomeric surfactants to exhibit the unique properties.

The excellent properties of the oligomeric cationic surfactants are attributed to their specific molecular structures. The hydrophilic head groups in the oligomeric cationic surfactant molecules are linked with each other by chemical bonds. These connections lead to strong interactions among the hydrocarbon groups as well as enhanced hydrophobic association. Besides, the repulsion among the hydrophilic head groups is also markedly weakened owing to the chemical bonding. Thus, compared with the traditional surfactants, the oligomeric cationic surfactants possess a higher surface activity and viscosity-enhancing ability and are more likely to form the wormlike micelles [16]. Overall, the oligomer surfactants reveal a significant application potential in tertiary oil recovery [6] due to the following advantages [17]: (1) their low critical micelle concentration (CMC) and high surface activity enable them to exhibit an ultra-low interfacial tension at a relatively low concentration [18], thus significantly improving the oil displacement efficiency; (2) the Gemini surfactants possess superior water solubility due to the presence of two hydrophilic head groups in their molecular structures [19]; (3) the concentration of the Gemini surfactant required for solubilizing the same amount of crude oil is much lower than that of the traditional surfactants. The requirements of a low concentration have a potential significance for the microemulsion-based oil displacement technology; (4) the Gemini surfactants have superior compatibility with the other oil displacement additives [20], thus the actual application costs can be reduced; (5) since each Gemini surfactant contains two hydrophilic head groups, these surfactants demonstrate enhanced salt tolerance as compared with the traditional surfactants.

Recently, the CO_2_-responsive switchable surfactants have emerged as a research hotspot in the field of colloidal chemistry [21,22]. The long-chain tertiary amines can be protonated by bubbling CO_2_ in the aqueous solutions of these surfactants [7]. In the protonated state, the long-chain tertiary amines can behave as surfactants owing to their structure comprising of a charged hydrophilic head and a long hydrophobic tail. Therefore, such surfactants can be used as stabilizers in the emulsion polymerization [23]. The surfactant can be “turned off” simply by bubbling the non-acid gases such as nitrogen, argon or air in the emulsion. The advantage of this method is that the properties of the aqueous solution, including emulsification and viscosity [24], can be altered by the introduction or removal of CO_2_. This is expected to be beneficial to the production of crude oil. The emulsification property is conducive to crude oil dispersion in water; high viscosity is useful for displacement of underground crude oil; the CO_2_-switching property is helpful in controllable post-treatment after oil production.

The oligomeric surfactants with the CO_2_-triggered switching properties have not been reported so far. In this study, a series of the CO_2_-responsive switchable surfactants from epoxidized soybean oil were prepared. The influence of CO_2_ on the surfactant performance was investigated by comparing the surface activity, viscosity, and other properties of their aqueous solutions.

## 2. Results and Discussion

### 2.1. Molecular Structure and CO_2_-Switching

The molecular structure of the novel surfactant is presented in Scheme 1. The oligomer contains a tertiary amine which can theoretically react with carbon dioxide reversibly. The reaction between carbon dioxide and tertiary amine leads to the bicarbonate formation. Heating or bubbling an inert gas into the oligomer solution can reversibly transform the bicarbonate to the primary amine state. Scheme 2 demonstrates the mechanism of the reversible reaction between the tertiary amine and CO_2._ The pK_aH_ values (pK_a_ of the conjugate acids) of oligomeric surfactant were obtained by theoretical calculation [25] and determined to be from 8.37 and 9.84. When the pK_aH_ value of an amine is in the range of 7–12, the reaction with CO_2_ can usually be easily reversed. In the presence of CO_2_ dissolved in the aqueous solution, the tertiary amine in the oligomer surfactant can undergo reversible protonation and becomes positively charged.

Figure 1 shows the ^1^H-NMR spectrum of the oligomeric surfactant. A peak of the terminal -CH_3_ group appeared at 0.80 ppm, and the chemical shift of the long alkyl chains was found between 1.40 and 1.63 ppm. The NMR signal with a chemical shift of 3.64–3.93 ppm was attributed to the methyl proton of the hydroxyl groups. Chemical shifts of 1.52, 2.54 and 2.82 ppm were attributed to the tertiary amine groups, which proves the ring opening of epoxy groups. The location of the primary amine-attacking epoxy ring is not definite, so the resulting surfactant product should be a mixture of several chemicals having similar molecular structures. The mass and ^13^C NMR spectra (Appendix A) also prove the synthesis of the oligomeric surfactant.

The conductivity of the aqueous solutions containing the oligomeric surfactant (1 mmol/L) in the two switchable cycling was realized by bubbling CO_2_ and N_2_ into the solutions at 25 ± 0.5 °C. The continuous injection of CO_2_ was intended to achieve the saturated dissolution of CO_2_ in the solution. Bubbling N_2_ into the solution guaranteed the removal of CO_2_ and subsequently accelerated the neutralization of the carbonate salts. The conductivity values after alternating bubbling of CO_2_ and N_2_ are presented in Figure 2. After bubbling CO_2_ in the solution for 20 min, the conductivity of the oligomer solution was observed to improve from 0.3 to 405 µS/cm. The conductivity returned to the original level after bubbling N_2_ into the solution. The oligomer molecules which had not undergone protonation were converted to the carbamate salts in the presence of water and CO_2_. Apparently, the solutions of oligomeric surfactant exhibited considerable switching behavior due to the presence of the tertiary amines in the molecular structure.

### 2.2. Surface Activity

Surface activity is one important index for evaluating surfactants. It can be observed from Figure 3a that the surface tension decreased sharply on enhancing the surfactant concentration. However, once the surfactant concentration reached the CMC value (the inflection points at which the plateau phase appeared), the surface tension no longer declined on increasing the surfactant concentration and entered a plateau phase, and the surface tension at CMC is treated as the lowest surface tension (*γ*_*CMC*_). As can be seen from Figure 3a, the oligomeric surfactant is an effective CO_2_ switchable surfactant; in the absence of CO_2_, it is a hydrophobic chemical. Due to the solubility limit, the surface tension curve has no break point, which means that oligomeric surfactant has poor surface activity without CO_2_. In the presence of CO_2_, the amine groups of the surfactant are protonated and this increases the surface activity of the solution, resulting in the decrease in surface tension. With the help of CO_2_, the CMC of the oligomeric surfactant is 0.2 mmol/L and *γ*_*CMC*_ is 35.8 mN/m. After sparging N_2_ into the aqueous solution to remove CO_2_, the surface tension value of the solution almost returns to the original value of pure water. Therefore, oligomeric surfactant is a type of CO_2_ switchable surfactants.

Expectedly, CO_2_ can lead to a significant change in the surface tension of the aqueous solution of the oligomeric surfactants. In addition, the behavior of the surface tension towards CO_2_ is reversible. The surface tension of the aqueous solution containing the oligomer was measured at 25 ± 0.5 °C after two alternating CO_2_ and N_2_ bubbling processes (Figure 3b). N_2_ was injected in the solution to promote the removal of CO_2_ and subsequently neutralize the ammonia salt. Figure 3b shows the shifting trends of the surface tension and pH of the solution during the two alternating CO_2_ and N_2_ bubbling cycles. After bubbling CO_2_ in the solution for 25 min, the surface tension was noted to decrease from 66.1 to 35.8 mN/m. In the CO_2_/N_2_ bubbling cycle, the pH value varied from 7.1 to 5.2. Consequently, after bubbling CO_2_ in the aqueous solution, its pH returned to the original value.

### 2.3. Viscosity-Enhancing Ability

The oligomeric surfactant exhibits a specific viscosity-enhancing behavior. The relationship between its apparent viscosity (η_app_) and the shear rate ($\dot{\gamma }$) at gradient concentrations has been presented in Figure 4.

As can be observed from Figure 5a, the oligomeric surfactant exhibited a strong viscosity-enhancing ability. At a low concentration, it failed to exhibit a distinct viscosity-enhancing effect. However, a relatively distinct viscosity-enhancing ability was observed as the concentration of the surfactant exceeded 6 mmol/L. At this stage, as the shear rate was increased further, the apparent viscosity of the solution remained relatively stable. The observed difference might be attributed to an enhanced number of hydrophobic chains in the surfactant molecule, which strengthened the hydrophobic interaction among the molecules in the solution and relatively stabilized the microstructure of the surfactant, thereby ensuring that the spatial net-like structure of the surfactant was not destroyed under high shear rates.

As the shear rate was increased, the data points were observed to be evenly distributed, thereby forming a relatively smooth plateau phase (Figure 4). Extrapolating the shear rate along the direction of concentration increase within the plateau phase to “0”, the zero-shear viscosities at different concentrations (η_0_) were obtained. The zero-shear viscosity can be perceived as the viscosity of the surfactant solution without any external force acting on the system. The relation between η_0_ and C was obtained by plotting the zero-shear viscosity as a function of the concentration of the surfactant. Specifically, Figure 5a compares the relation between the zero-shear viscosity and surfactant concentration for the oligomeric surfactant.

As can be seen from Figure 5a, varying the concentration of the surfactant solution within the designed concentration range lower than 6.0 mmol/L, the zero-shear viscosities of the solutions were very close to that of water (about 1 mPa·s), and the solutions exhibited almost no viscosity-enhancing ability. As the concentration of the solution ranged from 6.0 to 10.0 mmol/L, the solution viscosities were observed to increase, demonstrating a certain viscosity-enhancing effect. However, the effect was not significant. The observed minor increment could be attributed to the fact that at a relatively low concentration, the micelles formed by the surfactant did not entangle with each other and formed a net-like structure, indicating that the spatial structure formed by the micelles was unstable. As the solution concentration exceeded 12.0 mmol/L, the apparent viscosity (η_app_) of the surfactant solutions increased significantly with the surfactant concentration. This phenomenon could be attributed to the rapid growth of micelles and the formation of the semi-dilute solutions. The worm-like micelles in microscopic view was formed from Cryo-TEM of the oligomeric surfactant (Figure 3), entangling with each other due to the concentration exceeding the concentration of the sub-concentrated solution, thus, resulting in an enhanced solution viscosity.

It can be suggested that the difference in the number of hydrophobic chains and hydrophilic head groups correspondingly affected the morphology of the micelles formed by the surfactants in the solutions. The higher the number of hydrophobic chains, the stronger the hydrophobic interaction among the surfactant molecules. Likewise, the higher the number of hydrophilic head groups, the stronger the interaction between the surfactant and water or the hydration effect. These changes ensured a compact spatial net-like structure formed by the micelles in the aqueous solution, thus leading to an effective viscosity-enhancing performance.

Similarly, CO_2_ can trigger significant changes in the viscosity of the aqueous solution containing the oligomer, and the viscosity behavior towards CO_2_ is reversible. The viscosity of the aqueous solution containing the oligomer (12.0 mmol/L) was measured at 25 ± 0.5 °C after three alternating CO_2_ and N_2_ bubbling processes. N_2_ was injected in the solution to promote the removal of CO_2_ from the solution and subsequently neutralize the ammonium salt. Figure 5b demonstrates the trends of the solution viscosity during the two alternating CO_2_ and N_2_ bubbling cycles. After bubbling CO_2_ in the solution for 20 min, the viscosity was observed to increase from 1 to 320 mPa·s. After the removal of CO_2_ from the aqueous solution, the viscosity returned to the original value. Thus, the solution viscosity can be regulated by bubbling or removing CO_2_.

## 3. Materials and Methods

### 3.1. Materials

Epoxidized soybean oil and 3-dimethylaminopropylamine were obtained from Adamas Pharmaceuticals, Inc. (Emeryville, CA, USA). CO_2_ (≥99.998%) and N_2_ (99.998%) were supported by Xuyuan Chemical Industry Co. Ltd. (Chengdu, China) and used as received. Ultrapure water with a conductivity of 18.25 μS·cm^−1^ was prepared by an ultrapure water purification system (Chengdu Ultrapure Technology Co., Ltd., Chengdu, China).

### 3.2. Synthesis

Epoxidized soybean oil (9.19 g, 0.1 mol) and 3-dimethylaminopropylamine (6.12 g, 0.6 mol) and acetonitrile (60 mL) added to a three-neck distillation flask equipped with a reflux condenser. The reaction was continued at 80 °C in an N_2_ atmosphere under magnetic stirring for 10–15 h. After cooling to room temperature, the product was first dissolved in acetone and then poured into diethyl ether to precipitate. After separating the precipitate, this step was repeated three times to obtain the purified oligomeric surfactant. The experimental process is shown in Scheme 1. The synthetic route for preparing the surfactants is shown in Scheme 1.

### 3.3. Performance Analysis

The conductivity of aqueous solutions was measured with an EF30 conductometer (Mettler Toledo, Mississauga, Canada) at 25 °C while bubbling CO_2_ or N_2_ alternately.

A series of aqueous solutions with gradient concentrations of the surfactants to be tested were prepared by dissolving the surfactants in a specific amount of tri-distilled water. The surface tension (γ) values (mean of three replicates) of the aqueous solutions were determined by following the platinum plating method. The curves of the surface tension as a function of the surfactant concentration were plotted. A clear reduction in surface tension with increasing concentration was observed, followed by an obvious break point regarded as the CMC; the surface tension at the CMC was treated as the lowest surface tension (*γ*_*CMC*_).

The aqueous solutions with gradient concentrations of the surfactants were prepared by dissolving the surfactants in a specific amount of tri-distilled water. The apparent viscosity of the solutions was measured with Physica MCR301 rheometer (Anton Paar GmbH, Graz, Austria). The curves of the apparent viscosity (η_app_) as a function of the shear rate were plotted.

The ^1^H NMR and ^13^C NMR spectra were obtained on a Bruker Avance-II 400 MHz NMR spectrometer. Mass spectrometry analyses were performed on an HP 5890 Series II gas chromatograph (Hewlett-Packard, Les Ulis, France). The specimen was transferred to a Tecnai G2 F20 cryo-microscope (FEI Company, Hillsboro, OR, USA) using Gatan 626 cryo-holder and its workstation. The acceleration voltage was 200 kV, and the working temperature was kept below −170 °C.

## 4. Conclusions

In this study, novel CO_2_-responsive oligomeric surfactants have been successfully synthesized from epoxidized soybean oil. The molecular structure of the surfactants was verified through structural analysis. The oligomeric surfactants exhibited high surface activity after bubbling CO_2_ in the solutions. In presence of CO_2_, the CMC of the oligomeric surfactant is 0.2 mmol/L. For the sake of CO_2_, the viscosity of the aqueous solution can be changed between 1.0 and 400 mPa·s. For the aqueous solutions of the surfactants with same concentration, the viscosity-enhancing ability was significantly enhanced with increased concentration. In the same way, the surface activity of the aqueous solutions of the surfactants could be changed periodically by the introduction and removal of CO_2_, and the surface tension can be reversibly changed from 66.1 to 35.8 mN/m.

## Data Availability

The data presented in this study are available on request from the corresponding author.

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
