# Peer review of "Soybean-Oil-Based CO2-Switchable Surfactants with Multiple Heads"

_molecules, 2021, doi:10.3390/molecules26144342_

Round 1
Reviewer 1 Report
Manuscript Title: Soybean-oil-based CO2-switchbale Surfactants with Multiple Heads
Journal Title: Molecules
Authors: Huiyu Huang, Xiaoling Huang, Hongping Quan and Xin Su
Corresponding author: Xin Su
In this manuscript the authors prepared the oligomeric surfactants using epoxidized soybean-oil. Prepared surfactant has CO2-triggered switching properties. Prepared surfactant was investigated by advanced methods such as NMR spectroscopy. The performance of the CO2- triggered switching surfactant was studied by conductivity, surface tension value, viscosity. This work will contribute and will be of interest to the reader of the Molecules, topic of the manuscript is relevant. Based on these points, this manuscript can be recommended for publication in the journal.
-it is recommended to add real pictures that can show CO2-triggered switching properties of the surfactant.
Author Response
In this manuscript the authors prepared the oligomeric surfactants using epoxidized soybean-oil. Prepared surfactant has CO2-triggered switching properties. Prepared surfactant was investigated by advanced methods such as NMR spectroscopy. The performance of the CO2- triggered switching surfactant was studied by conductivity, surface tension value, viscosity. This work will contribute and will be of interest to the reader of the Molecules, topic of the manuscript is relevant. Based on these points, this manuscript can be recommended for publication in the journal.
Reply (R): Sincerely thank the reviewer for his/her deep understanding and positive comments on our work.
-it is recommended to add real pictures that can show CO2-triggered switching properties of the surfactant.
R1: Thank the reviewer for the suggestion. Figure 5 (c) was newly added to demonstrate the appearance change of surfactant aqueous solution before and after bubbling CO2.
Reviewer 2 Report
Manuscript ID: molecules-1282813 Title: Soybean-oil-based
CO2-switchbale Surfactants with Multiple Heads Authors: Huiyu Huang,
Xiaoling Huang, Hongping Quan, Xin Su *
This work is about CO2-switchable oligomeric surfactants displaying novel properties that has bridged the gap between traditional and polymeric surfactants. They present high efficiency, multifunctionality and environmental friendliness. The authors present the 1H NMR spectrum to prove that the oligomeric surfactant was synthetized. This one contains a tertiary amine which can react with carbon dioxide reversibly. The reaction between carbon dioxide and tertiary amine leads to the bicarbonate formation. Heating or bubbling an inert gas into the oligomer solution can reversibly transform the bicarbonate to the primary amine state. The conductivity values after bubbling of carbon dioxide is improved, and the conductivity returns after bubbling with nitrogen. In the presence of carbon dioxide the surface tension decreases and after sparging nitrogen to remove carbon dioxide, the surface tension returns to the original value of pure water. The viscosity was also determined. The oligomeric surfactant exhibits a strong viscosity-enhancing ability.
This study presents the synthesis of an oligomeric surfactant and makes its characterization using the mentioned techniques. The text is clear and very well presented. The experimental procedures are complete and clearly described. References are updated. I recommend it to be published in this journal
Author Response
Reply: We appreciate the reviewer for his/her positive comments.
Reviewer 3 Report
CO2-switchable surfactants represent an interesting class of surfactants addressed in this manuscript. Some points should be clarified before pubblication of the manuscript.
Title: typo mistake "switchable"
Introduction: The potential applications for CO2-switchable surfactants should be more detailed.
Results and discussion: C-NMR and mass analysis of the synthesized oligomeric surfactant should be provided, at least. Otherwise, it is not correct to affirm in the conclusion that "in-depth structural analysis" was performed.
The CMC of the surfactans under CO2 should be determined from surface tension data.
Paragraph 2.3 is redundant in the description of the viscosity-enhancing ability of this surfactant and it should be reorganised and rewritten. The terms tetrameric and oligomeric are using in a confusing way referring to the synthesized surfactant. In figure 5a, is it correct in the legend "under air" or is better "under N2"?
Line 180 Figure 7 is missing in the manuscript
Line 193-203 the discussion considers the formation of micelles and worm-like micelles but there is no evidence about that in the manuscript.
Methods:
Line 242-243 It is reported conductivity analysis but no results are presented in the manuscript.
Conclusions:
Line 256-262 The sentences are not clear
Author Response
Responses to Reviewers Comments on Manuscript molecules-1282813
Responses to Reviewer 3:
Comments to the Author:
CO2-switchable surfactants represent an interesting class of surfactants addressed in this manuscript. Some points should be clarified before publication of the manuscript.
Reply (R): Thank the reviewer for providing comments and suggestions.
Title: typo mistake "switchable"
R1: Thank the reviewer for his/her kind reminding. We check the spelling of the title and the wrong word was corrected as “switchable”.
Introduction: The potential applications for CO2-switchable surfactants should be more detailed.
R2: The expression was added in Introduction, as “This is expected to be beneficial to production of crude oil. The emulsification property is conducive to crude oil dispersion in water; high viscosity is useful to displacement of underground crude oil; CO2 switching property is helpful to controllable post-treatment after oil production.”
Results and discussion: C-NMR and mass analysis of the synthesized oligomeric surfactant should be provided, at least. Otherwise, it is not correct to affirm in the conclusion that "in-depth structural analysis" was performed.
R3: The mass and 13C NMR spectra was attached in Supporting Information (SI). As the oligomeric surfactant is a kind of mixture, it is hard to get helpful information from the mass and 13C NMR spectra. Therefore, the expression “in-depth” was deleted.
The CMC of the surfactants under CO2 should be determined from surface tension data.
R4: The expression was added in section 2.2, as “With the help of CO2, the CMC of the oligomeric surfactant is 0.2 mmol/L”.
Paragraph 2.3 is redundant in the description of the viscosity-enhancing ability of this surfactant and it should be reorganised and rewritten. The terms tetrameric and oligomeric are using in a confusing way referring to the synthesized surfactant. In figure 5a, is it correct in the legend "under air" or is better "under N2"?
R5: In section 2.3, the word “tetrameric” was changed into “oligomeric”; the sentences “The viscosity-enhancing effect observed for the tetrameric, trimeric and Gemini surfactants clearly demonstrated that at the same concentration, the tetrameric surfactant exhibited the strongest viscosity-enhancing behavior, followed by the trimeric and Gemini surfactants.” was deleted to avoid misunderstanding.
Moreover, in Figure 5a, the legend “under air” was modified as “under N2”.
Line 180 Figure 7 is missing in the manuscript
R6: Thank you for pointing out the issue. In the sentence “As can be seen from Figure 7, varying the concentration of the surfactant solution”, Figure 7 was modified as Figure 5a.
Line 193-203 the discussion considers the formation of micelles and worm-like micelles but there is no evidence about that in the manuscript.
R7: Cryo-TEM image was supplied in SI. It can be observed that worm-like micelles are formed after bubbling CO2.
Methods:
Line 242-243 It is reported conductivity analysis but no results are presented in the manuscript.
R8: Figure 2 shows the conductivity values of the aqueous solution with the oligomeric surfactant (0.5 wt.%) after alternating bubbling of CO2 and N2 at 25 ± 0.5 °C. The conductivity analysis was used to obtain data for Figure 2.
Conclusions:
Line 256-262 The sentences are not clear
R9: In conclusion, we added more details as “In this study, novel CO2-responsive oligomeric surfactants have been successfully synthesized from epoxidized soybean oil. The molecular structure of the surfactants was verified through structural analysis. The oligomeric surfactants exhibited high surface activity after bubbling CO2 in the solutions. In presence of CO2, the CMC of the oligomeric surfactant is 0.2 mmol/L. For the sake of CO2, the viscosity of the aqueous solution can be changed between 1.0 and 400 mPa·s. For the aqueous solutions of the surfactants with same concentration, the viscosity-enhancing ability was significantly enhanced with increased concentration. In the same way, the surface activity of the aqueous solutions of the surfactants could be changed periodically by the introduction and removal of CO2, and the surface tension can be reversibly changed from 66.1 to 34.8 mN/m.”
Other modification:
Scheme 1 was modified: In Scheme 1, it should be 4 rather than 3.
Supporting information was also supplied and it contains the images and data obtain through Cryo-TEM, mass and 13C NMR spectra.
In section 3.2, the description was added: “The 1H NMR and 13C NMR spectra were obtained on a Bruker Avance-II 400 MHz NMR spectrometer. Mass spectrometry analyses were performed on an HP 5890 Series II gas chromatograph (Hewlett-Packard, Les Ulis, France). The specimen was transferred to a Tecnai G2 F20 cryo-microscope (Hillsboro, Oregon, USA) using Gatan 626 cryo-holder and its workstation. The acceleration voltage was 200 kV, and the working temperature was kept below −170 â—¦C.”

Round 2
Reviewer 3 Report
The CMC of the surfactants under CO2 should be determined from surface tension data.
R4: The expression was added in section 2.2, as “With the help of CO2, the CMC of the oligomeric surfactant is 0.2 mmol/L”.
The authors should add in the materials section (line 260-261) how "The corresponding values of CMC and ???? were extracted from the curves" For instance segmental fitting, straight line intercept or others.
Author Response
Reply: Thank you for the suggestion. Figure 3a was modified. The auxiliary lines and arrows were added in Figure 3a to show the values of CMC and ????.
Meanwhile, the expression at Line 142 was changed as “With the help of CO2, the CMC of the oligomeric surfactant is 0.2 mmol/L and ???? is 35.8 mN/m.”
The description was added at line 265, as “A clear reduction in surface tension with increasing concentration was observed, followed by an obvious break point regarded as the CMC; the surface tension at the CMC was treated as the lowest surface tension (????).”
